# Optical spin-orbit torque in heavy metal-ferromagnet heterostructures

Gyung-Min Choi [1,2,3,8]✉, Jung Hyun Oh[4,8], Dong-Kyu Lee[4], Seo-Won Lee[4], Kun Woo Kim [5], Mijin Lim[6], Byoung-Chul Min [3], Kyung-Jin Lee [4,7]✉ & Hyun-Woo Lee [6]✉

Spin current generation through the spin-orbit interaction in non-magnetic materials lies at the heart of spintronics. When the generated spin current is injected to a ferromagnet, it produces spin-orbit torque and manipulates magnetization efficiently. Optically generated spin currents are expected to be superior to their electrical counterparts in terms of the manipulation speed. Here we report optical spin-orbit torques in heavy metal/ferromagnet heterostructures. The strong spin-orbit coupling of heavy metals induces photo-excited carriers to be spin-polarized, and their transport from heavy metals to ferromagnets induces a torque on magnetization. Our results demonstrate that heavy metals can generate spin-orbit torque not only electrically but also optically.

[1] Department of Energy Science, Sungkyunkwan University, Suwon 16419, Korea. [2] Center for Integrated Nanostructure Physics, Institute for Basic Science (IBS), Suwon 16419, Korea. [3] Center for Spintronics, Korea Institute of Science and Technology, Seoul 02972, Korea. [4] Department of Materials Science and Engineering, Korea University, Seoul 02841, Korea. [5] Center for Theoretical Physics of Complex Systems, Institute for Basic Science (IBS), Daejeon 34051, Korea. [6] Department of Physics, Pohang University of Science and Technology, Pohang 37673, Korea. [7] KU-KIST Graduate School of Converging Science and Technology, Korea University, Seoul 02841, Korea. [8] These authors contributed equally: Gyung-Min Choi, Jung Hyun Oh. ✉email: gmchoi@skku.edu; kj_lee@korea.ac.kr; hwl@postech.ac.kr

When a spin current polarized along the direction **p** is injected into a ferromagnet (FM) with magnetization **M**, it produces a spin torque[1,2] along the direction **M**×(**M** × **p**), which is a powerful tool to manipulate the magnetization of FM. There are ongoing searches for ways to generate a more powerful spin torque. One option is to electrically generate a spin current from the spin–orbit coupling of a non-magnetic material[3–5]. Such a spin torque is called a spin–orbit torque, and the electrical generation of a spin–orbit torque through the spin Hall effect in heavy metals (HM) is intensively studied. For the fast magnetization dynamics, however, the optical generation of a spin–orbit torque is more promising than its electrical counterpart.

A prerequisite for the optical generation is the photo-spin conversion, which has been intensively studied for semiconductors such as GaAs[6–8]. A well-known mechanism of the conversion is the optical orientation; photo-excitation by a circularly polarized light with angular momentum along the direction **σ** can generate excited electrons that are spin-polarized along the direction **σ**. Such optically generated spins have been used to generate a spin torque in a ferromagnetic semiconductor MnGaAs[9]: Spin-polarized electrons in GaAs produced by optical orientation interact with magnetic moments of Mn to generate a spin torque along the direction **M**×(**M** × **σ**). An alternative method to optically generate a spin torque has been demonstrated for MnGaAs[10]. Its basic idea is to modify the magnetocrystalline anisotropy by light, thereby creating an anisotropy-induced torque. Although this torque is named optical spin–orbit torque[10], it is independent of the angular momentum **σ** of light and does not involve photo-spin conversion. Thus it is not the exact optical counterpart of the electrical spin–orbit torque. All these works with ferromagnetic semiconductors are limited by the low Curie temperature of the materials. Still, another possible option is to optically generate a spin current in two-dimensional electron gas via circular photo-galvanic effect[11] or spin-galvanic effect[12] and inject it to a neighboring metallic FM with high Curie temperature. However, the resulting spin current flows within the electron gas and its injection into a FM is rather difficult.

The optical torque on magnetization has been demonstrated at room temperature for magnetic materials such as ferrimagnetic insulator, ferrimagnetic metal, and ferromagnetic metal[13–16]. The torque in these materials is qualitatively different, however, from the optical-orientation-induced torque illustrated for magnetic semiconductors[9]; the former torque points along the direction **M**× **σ**, which contrasts with the latter torque that points along the direction **M**× (**M** × **σ**). Thus the former torque has been attributed to a different mechanism called inverse Faraday effect[17]. We call the former and the latter torques field-like and damping-like torques, respectively, since they have similar structures as the field-like and damping-like components of electrically generated spin–orbit torque[18]. Interestingly some of us showed that even ferromagnetic metals such as Fe, Co, and Ni can exhibit a sizable damping-like torque in certain situations[16]: whereas light incident on those FMs produces mostly a field-like torque when the FMs are capped by a thin MgO or Au layer, a damping-like torque becomes comparable to a field-like torque in magnitude when the FMs are capped by Pt. Roles of Pt were not clarified, however.

In this work, we clarify the crucial role of Pt and show for a Pt/Co bilayer that a damping-like torque can even dominate over a field-like torque provided the Pt layer is sufficiently thick. Moreover, we demonstrate that not only Pt, but also various HM produce similar results in HM/Co bilayers. Our analysis indicates that the damping-like torque in HM/Co bilayers is caused by efficient optical spin orientation in HMs through their strong spin–orbit coupling. These results resemble, in many ways, the

electrically generated spin–orbit torque that also utilizes the strong spin–orbit coupling of HMs[18]. We thus call the optically generated damping-like torque in HM/Co layers as optical spin–orbit torque (OSOT).

## Results

**Experiment.** We study samples of the film structures of sap/Pt ($d_{Pt}$)/Co(3)/MgO(3), sap/Pt(5)/Cu($d_{Cu}$)/Co(3)/MgO(3), and sap/Co($d_{Co}$)/Pt(2), where the sap is the sapphire substrate with (0001) texture, and $d_X$ is the thickness of the layer X in nanometers. On top of the MgO capping, we deposit an additional Ta 1 nm to secure the passivation of MgO. We also study samples with other HM of Ta, W, or Pd instead of Pt. All layers are deposited by magnetron sputtering with a base pressure of $<5 \times 10^{-8}$ Torr without breaking the vacuum. To generate and detect the OSOT, we use a time-resolved pump-probe technique (see Methods). To generate the OSOT, a circularly polarized pump light with the photon energy of 1.58 eV and pulse width of 1.1 ps is incident on the sapphire substrate side of the samples: for sap/Pt/Co or sap/Pt/Cu/Co samples, the pump passes the substrate layer and then the Pt layer; for sap/Co/Pt samples, the pump passes the substrate and Co layer and then the Pt layer. Since photons with right circular polarization (RCP) and left circular polarization (LCP) carry the angular momentum of $+\hbar$ and $-\hbar$, respectively (For the light helicity, we adopt the handedness convention from the point of view of the source.), circularly polarized photons can serve as an angular momentum source for the OSOT. To detect the OSOT, a linearly polarized probe light is incident on the surface side of the samples to detect the dynamics of the $z$ or $y$-component of magnetization by the polar or longitudinal magneto-optical Kerr effect (MOKE) (see Methods). The magnetization of Co is initially saturated to the $x$-direction by an external magnetic field, and the pump light propagates along the $z$-direction (Fig. 1). All experiments are performed at room temperature and incident pump fluence of 10 J m$^{-2}$. The measured signals are linearly proportional to the fluence (Supplementary Note 1 and Supplementary Fig. 1).

**Optical spin–orbit torque.** We first show the OSOT in the structure of sap/Pt(5)/Co(3)/MgO(3). Figure 2a shows the resultant polar MOKE signal oscillation of this structure right after the pump pulse is shot at time zero. With a polar MOKE, the Kerr rotation measures $M_z$ dynamics of Co (Supplementary Note 2 and Supplementary Fig. 2). The oscillation changes its sign for an opposite circular polarization of the pump and disappears for a linear polarization of the pump. (We align the in-plane crystal axis of Co along the applied magnetic field so that a torque due to a sudden change of magnetic anisotropy is suppressed.) The pump polarization dependence evidences a critical role of the

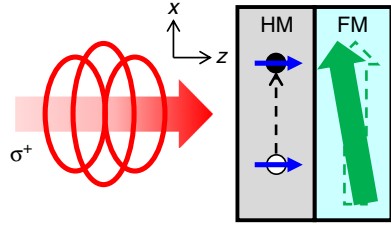

**Fig. 1 Schematics for mechanism.** A right circularly polarized light (σ⁺) generates a spin-polarized electron (filled circle) and hole (empty circle) excitations in HM (Blue arrows indicate the spin angular momentum). A spin transport from HM to FM induces a damping-like torque on the magnetization of FM (green arrow). A light propagates to the z-direction, and initial magnetization lies along the x-direction.

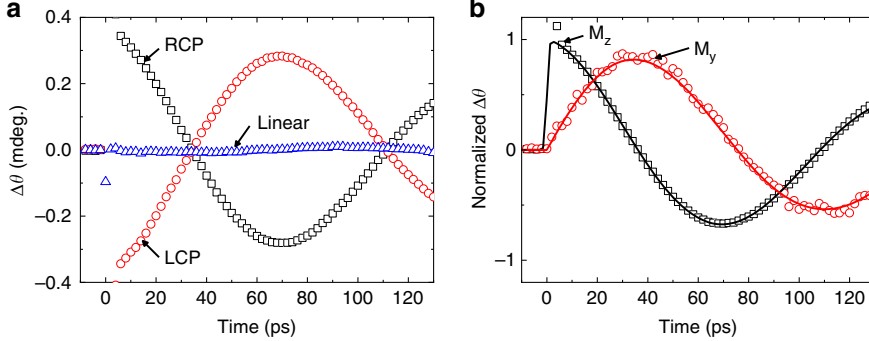

**Fig. 2 Optical spin–orbit torque. a** The measured dynamics of the *z*-component of Co magnetization in the sap/Pt(5)/Co(3)/MgO(3) sample driven by pump pulse, incident on the substrate side, with right circular polarization (RCP) (black squares), left circular polarization (LCP) (red circles), and linear polarization (blue triangles). Given the negative value of the static Kerr rotation of Co (Supplementary Note 6), positive/negative $\Delta\theta$ of **a** indicates $-z/+z$ tilting of magnetization. **b** The comparison of the $\mathbf{M}_z$ (black squares) and $\mathbf{M}_y$ (red circles) dynamics of Co in the sap/Pt(5)/Co(3)/MgO(3) sample. The black and red lines are fitting with cosine and sine functions, respectively.

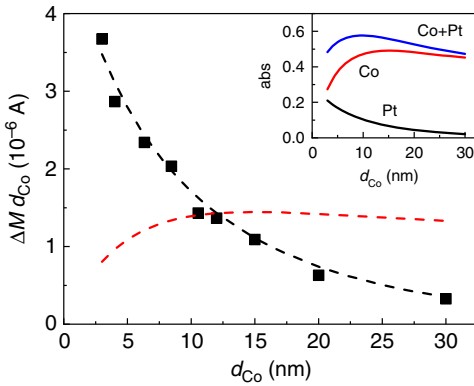

**Fig. 3 Comparison with light absorption. a** The amount of measured optical spin–orbit torque in the sap/Co($d_{Co}$)/Pt(2) structure (black squares). Dashed lines are fittings with light absorption in Pt (black line) and in Co (red line). For the fitting, we use a quantum efficiency ($\eta$) of 0.045 and 0.008 in Eq. (1) for black and red lines, respectively. Inset shows the calculation of light absorption (abs) of Pt (black line), Co (red line), and Co+Pt (blue line) in the sap/Co($d_{Co}$)/Pt(2) structure (light is incident on the sapphire substrate side). The calculation is done with refractive indexes of each layer and initial light energy of one.

pump light's angular momentum in the OSOT. The oscillation period is about two orders of magnitude longer than the pump pulse width. Thus after the initial tilting of **M** by the pump, the subsequent magnetization dynamics is governed by the magnetic anisotropy of the structure. The oscillation frequency of 7 GHz agrees with the structure's ferromagnetic resonance frequency determined by the anisotropy. We also measure $M_y$ dynamics with a longitudinal MOKE (Supplementary Note 3 and Supplementary Figs. 3, 4). While $M_z$ dynamics exhibit a cosine-like oscillation, $M_y$ dynamics show a sine-like oscillation (Fig. 2b). Therefore, magnetization dynamics begins from the initial tilting towards the $\pm z$-direction, which agrees with the direction of the damping-like torque $\mathbf{M} \times (\mathbf{M} \times \boldsymbol{\sigma})$. The initial tilting towards the $\pm z$-direction demonstrates that the damping-like torque is dominant over the field-like torque $\mathbf{M} \times \boldsymbol{\sigma}$ that causes the initial tilting towards the $\pm y$-direction.

Discussion on the mechanism of the damping-like torque is in order. If the photo-spin conversion originates from a ferromagnetic order of FM or induced ferromagnetism at the HM/FM interface[19], the direction of the spin polarization should be determined by the direction of the magnetization rather than the

polarization of the light. We rule out this possibility because the direction of the initial tilting of magnetization is the same for the initial magnetization directions of $\pm x$ at polar MOKE geometry (Supplementary Note 3). If the inverse Faraday effect is dominant, the magnetization must initially tilt towards the *y*-direction since the circularly polarized pump pulse generates an effective magnetic field along the *z*-direction[13–16]. We also find that the phase of the oscillation is unaffected by inverting the layer stacking order from sap/Pt/Co to sap/Co/Pt (Supplementary Note 4 and Supplementary Fig. 5). This result rules out mechanisms based on the stacking sequence, such as temperature gradient.

An important clue for the mechanism is obtained from the dependence on the Co thickness $d_{Co}$. We measure the magnitude of the OSOT as a function of $d_{Co}$ from 3 nm to 30 nm, while fixing the Pt thickness to 2 nm in the sap/Co/Pt structure (Fig. 3). The variation of $d_{Co}$ modifies the relative weight of the pump light absorptions in Co and Pt. We calculate the light absorption in each layer using refractive indexes of each layer (Supplementary Note 5 and Supplementary Table I). We find that the pump absorption in Pt exhibits the exactly same $d_{Co}$ dependence as $\Delta M \cdot d_{Co}$, which is a measure of $\int J_s dt$, where $J_s$ is the spin (magnetic moment) current per area to Co. $J_s$ produces a torque on the magnetization of Co. In our experiment, $J_s$ appears as a short pulse, and the magnitude of magnetization dynamics is determined by $\int J_s dt$. The $\Delta M$ is determined experimentally from the relation $\Delta M = \frac{\Delta M}{M} \times M_s = \frac{\Delta\theta_{R-L}}{2\theta_K} \times M_s$, where $\Delta\theta_{R-L}$ is the Kerr rotation difference between RCP and LCP, $\theta_K$ is the Kerr rotation for the saturation magnetization (Supplementary Note 6 and Supplementary Fig. 6), and $M_s$ is the saturation magnetization of Co of $1.4 \times 10^6$ A$^{-1}$.

**Theoretical calculation.** Based on this experimental result, we propose the following mechanism: [Step 1: optical orientation] the pump light induces the optical dipole transition in Pt and generates electron-like and hole-like excitations above and below the Fermi energy, respectively (simply referred to as electrons and holes from now on), that are spin-polarized along $\pm z$-direction; [Step 2: spin current injection] the excited spins flow from Pt into Co to generate the OSOT. Since the injection of a spin current (polarized along $\pm z$-direction) into a FM produces a damping-like torque $\pm \mathbf{M} \times (\mathbf{M} \times \mathbf{z})$[18], this mechanism is consistent with all experimental data presented above. Note that in this mechanism, the pump light interacts with Pt, and there is no direct interaction between the light and FM. If the direct interaction were important, it would generate an effective magnetic field along

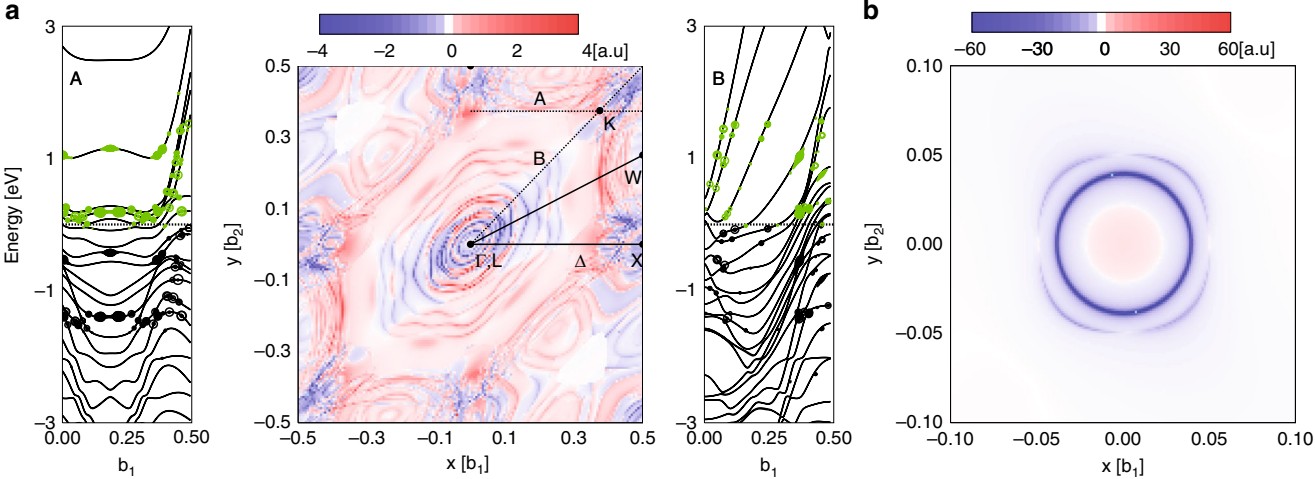

**Fig. 4 Theoretical calculation of optical orientation.** Optical spin generation rate ($ds_z/dt$) (positive value with red and negative value with blue) on the conduction band in **a** six layers of [111] Pt and **b** [111] GaAs in the **k** space with a continuous light of right-circular polarization ($+\hbar$). On the left and right panels of **a**, we plot energy levels along A and B lines indicated in the center panel and show associated optical transitions where the oscillation strength is proportional to the size of symbols (green symbols for electrons and black symbols for holes).

$\pm z$-direction, which results in a field-like torque $\pm \mathbf{M} \times \mathbf{z}$[16]. This demonstrates the importance of the Pt layer for the generation of the OSOT.

The optical orientation in [Step 1] demands further discussion. Although the optical orientation is well known in semiconductors such as GaAs, its applicability to HMs such as Pt may not be evident since these two classes of materials have very different band structures. In GaAs, $p$ orbitals in the valence bands make optical transitions to $s$ orbitals in the conduction bands, satisfying the electric dipole selection rule of $\Delta l = \pm 1$. As the total angular momentum $J$ is a good quantum number near the conduction band bottom and the valence band top, the resulting correlation between spin and orbital angular momenta allows polarized spins to be generated by the optical transition[6–8]. On the other hand, in the case of HMs such as Pt, the situation is more complicated since there are many energy bands near the Fermi level $\varepsilon_F$. However, there are similarities with GaAs, as well. For Pt, for instance, many energy bands below $\varepsilon_F$ have strong $d$ character, and there are energy bands above $\varepsilon_F$ that have mainly $p$ character. Thus, optical transitions between those bands satisfy the electric dipole selection rule $\Delta l = \pm 1$. Moreover, many energy eigenstates near $\varepsilon_F$ have well-defined correlations between spin and orbital angular momenta, although the total angular momentum $J$ is not a good quantum number. This escalates the similarity with GaAs further. As a passing remark, we mention that not only Pt but also many $4d$ and $5d$ transition metals share similar properties, and the spin–orbit correlation in those HMs is responsible for the Hund's rule type behavior of the intrinsic spin Hall effect in HMs[20].

To check whether the optical orientation is indeed possible in Pt, we calculate the evolution of spin density generated by light of the experimental condition (see Methods). In Fig. 4, we show the optical spin generation rate $ds_z/dt$ (positive value with red and negative value with blue) in the **k** space when a right-circularly polarized light is illuminated [(a) Pt and (b) GaAs]. For the photon energy of 1.58 eV, optical transitions occur at many **k** points (Fig. 4a) in Pt, which is natural considering that there are many energy bands near $\varepsilon_F$. This feature of Pt is in clear contrast to the case of semiconductors, where the contribution near the $\Gamma$ point dominates in general[21], which is consistent with our calculation for GaAs (Fig. 4b). Thus the situation is more

complicated in Pt than in GaAs: $ds_z/dt$ varies in magnitude and even in sign as a function of **k**. However, we emphasize that the **k** space region with positive $ds_z/dt$ is much wider than the **k** space region with negative $ds_z/dt$. This implies that despite the complication in Pt, there exists a clear preference towards one particular sign of $ds_z/dt$, confirming the optical orientation. By analyzing the eigenstates involved in the optical transition, we find that $p$ and $d$ orbitals of a Pt atom are important for those optical transitions. The summation over all **k** points determines the net spin-polarized carriers, which is non-zero. The ratio of the net spin $s_z$ generation rate of whole excitations (electrons and holes) to the photon absorption rate is calculated to be 0.11. Although a definite distinction of the spin polarization between electrons and holes is not possible due to the hybridization of $p$ and $d$ orbital, the sign of the spin polarization is the same for electrons and holes. On the contrary, GaAs has an opposite sign of the spin polarization for electrons and holes: the ratio of the spin generation rate to the photon absorption rate is $-0.436$ for electrons and 0.434 for holes at the photon energy of 1.58 eV. When the helicity of the light polarization is switched, the sign of the net spin-polarized carriers is reversed. This calculation supports the helicity-dependent spin generation in Pt (optical generation).

The spin current injection in [Step 2] also demands further discussion. The excited spins in HM can move to FM either by diffusive or drift transport. Due to the unbalance in the spin density, the optically generated spin excitations in Pt will diffuse to Co, generating a spin current (Fig. 5a). The efficiency of the spin diffusion depends on the spin diffusion length, which may be different for electrons and holes. Another possible mechanism for spin transport is the electric field ($E$-field) at the interface (Fig. 5b). The $E$-field induces a directional motion of the spin excitations, thus creating a drift spin current. Although the internal $E$-field at interfaces is studied mostly for semiconductor junctions, we expect it occurs also at metallic interfaces between metals with different work functions as a chemical equilibrium has to be achieved by transferring electrons at interfaces with a narrow width $\approx 1$ nm (screening length)[22,23]. Such an $E$-field is the key requirement for the Rashba spin splitting at metallic surfaces[24–26]. The direction of the drift current is determined by the work function difference and group velocity. In contrast to

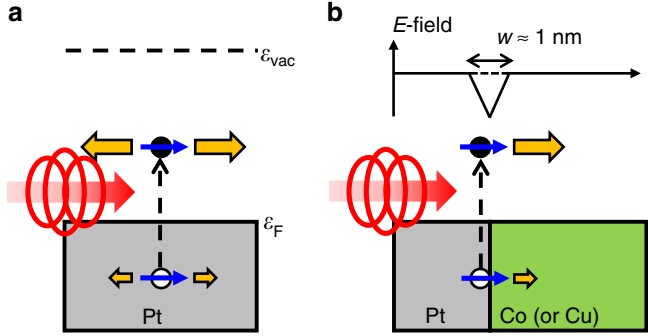

**Fig. 5 Mechanisms for spin transport. a** The mechanism based on the diffusion of electrons and holes. The optical orientation in Pt induces a spin polarization (blue arrows) on electrons in conduction state (filled circles)/ holes in valence state (empty circles). Both electrons and holes diffuse in all directions. The diffusivity (orange arrows) or diffusion length could be different for electrons and holes. $\varepsilon_F$ and $\varepsilon_{vac}$ represent the Fermi level and vacuum level, respectively. **b** The mechanism based on the drift of electrons and holes. An electric field (E-field) develops at the interface with a typical width (w) of ≈1 nm due to the difference in work functions. The sign of E-field is denoted as negative because the work function of Pt is higher than that of Co or Cu. The E-field induces a directional motion of electrons and holes at the interface. The drift velocity (orange arrows), determined by the summation of the group velocity over **k**-vectors, could be different for electrons and holes.

the semiconductor case, where electrons and holes have the group velocities in the opposite directions, electrons and holes in Pt do not have such anti-correlation in the group velocities. Depending on **k**-vectors, electrons and holes have the same or opposite sign of the group velocity (Fig. 4a). The average (integration over all **k**-vectors) direction tends to be the same for electrons and holes. Further study is necessary to figure out which of these mechanisms are more important.

**Quantum efficiency**. Next, we experimentally determine the quantum efficiency $\eta$ of the OSOT, which denotes the ratio between the number of spins transferred to Co and the number of photo-excited electrons generated in Pt. The number of spins transferred to Co is determined by $\Delta M \cdot d_{Co}/\mu_B$, where $\mu_B$ is the Bohr magneton. The number of photo-excited electrons in Pt is determined by $F_{in} \cdot A_{Pt}/\hbar\omega$, where $F_{in}$ is the incident pump fluence per pulse of $10\ \mathrm{J\,m^{-2}}$, $A_{Pt}$ is the light absorption probability in Pt, and $\hbar\omega$ is the photon energy. Then the quantum efficiency, $\eta$, for the OSOT is defined as,

$$\eta = \frac{\Delta M \cdot d_{Co}}{\mu_B} \div \frac{F_{in} \cdot A_{Pt}}{\hbar\omega} \qquad (1)$$

We find that the $\eta$ does not depend on Co thickness but does on Pt thickness $d_{Pt}$ (Fig. 6a), evidencing the critical role of Pt in the spin generation. The $\eta$ shows a maximum of ≈0.08 at $d_{Pt} = 2$ nm in the sap/Pt($d_{Pt}$)/Co(3)/MgO(3) structure, and it is somewhat smaller in the sap/Co(3)/Pt($d_{Pt}$) structure probably because the different morphology at the Pt/Co interface affects the spin transport efficiency from Pt to Co. The $\eta$ decreases gradually with increasing $d_{Pt}$, which is reasonable considering that as $d_{Pt}$ increases, spins should transfer longer distance before they get injected to Co. From the $d_{Pt}$ dependence of $\eta$, we obtain (see Methods) the spin diffusion length $l_s$ of Pt of 6 nm with the sap/ Pt/Co/MgO structure and 9 nm with the sap/Co/Pt structure, which are within the range of 1–10 nm of the previous report[27]. Since the OSOT is generated in two steps, we also attempt to assess the efficiency of each individual step: $\eta = P_s P_c$, where the spin polarization $P_s$ quantifies the efficiency of the optical

orientation in Pt [Step 1], and the collection probability $P_c$ quantifies the efficiency of the spin transport from Pt to Co [Step 2]. For the sap/Pt(2)/Co(3)/MgO(3) structure, where $\eta$ is maximized, our spin diffusion calculation with $l_s$ of Pt of 6 nm (see Methods) results in $P_c \approx 0.7$. The relation $\eta = P_s P_c$ then sets $P_s \approx 0.1$, which well matches the calculated ratio of the spin generation rate to the photon absorption rate of 0.11.

We also examine the bilayer structure variation. The OSOT persists even with the insertion of Cu between Pt and Co (Fig. 6b). This result indicates that direct contact between Pt and Co is not necessary for the OSOT. It also excludes the possibility that the magnetic proximity effect in Pt[19] is the source of the OSOT. The magnitude of the OSOT decreases with Cu thickness. From the spin diffusion simulation (see Method), we obtain the $l_s$ of Cu of ≈25 nm, which is much shorter than the reported $l_s$ of Cu, ≈400 nm[28]. One possible reason for the difference is that Jedema et al.[28]. probes the spin relaxation near the Fermi energy, whereas the spin excitations in our experiment can have considerably larger excitation energies of ≈1 eV. For such excitation energy, the spin–orbit coupling effect becomes strong even in Cu (in particular for holes), which can result in fast spin relaxation[29]. The spin Hall conductivity, which is a measure of the spin–orbit coupling strength, becomes sizable even for Cu at the excitation energy of a few eV[30] and comparable to that of the strong spin–orbit coupling material, Pt. Another possible reason is the coupling to charge transport. When there is an E-field at the Pt/Cu interface, a photo-excitation at certain **k**-vectors with a different sign of the group velocity between electrons and holes can induce a charge current. In this situation, the spin transport property could be different from that without charge current (Supplementary Note 7 and Supplementary Fig. 7). However, this estimation is based on the assumption that the E-field at the interface is strong enough to induce photocurrent, which has not been verified experimentally.

We find that the OSOT exists even when Pt is replaced by other HMs such as Ta, W, or Pd (Fig. 7), which implies that the OSOT is a common phenomenon in many HM/FM bilayers and HM/NM/FM trilayers, where NM is the non-magnetic metal of Cu. Although the magnitude of $\eta$ varies with HM, the sign of $\eta$ is the same for all HMs that we have examined, which is in contrast to the electrical spin–orbit torque whose sign varies with HM choices[5,31]. In addition, the magnitude of $\eta$ is relatively high for Pd and Pt, which have higher work functions than Co[32], and relatively small for Ta and W, which have lower work functions than Co[32]. We expect that the E-field at the HM/FM interface, driven by the work function difference, can affect the spin transport efficiency of electrons and holes at the HM/FM interface. Further experimental and theoretical efforts, especially for the magnitude and sign of the optical orientation in various HMs and effect of the work function difference on the spin transport, are needed to clarify the [Step 1] and [Step 2] of the OSOT mechanism.

Finally, we remark that we observe the OSOT not only at the photon energy of 1.58 eV as presented above but also at other photon energies such as 1.38 and 1.72 eV (Fig. 6a). For the Pt(6)/ Co(3)/MgO(3) sample, the value of $\eta$ at 1.72 eV is close to that at 1.58 eV, but the value of $\eta$ at 1.38 eV is about two times smaller than that at 1.58 eV.

In summary, we report the OSOT in the HM/FM hetero-structures. Our experimental and theoretical results suggest that the OSOT is driven by the spin excitation via optical orientation in HM followed by the spin transport from HM to FM. It is also demonstrated that the OSOT is a common phenomenon in HM/ FM bilayers and HM/NM/FM trilayers with various HMs. Further works are needed to clarify the detailed process of the OSOT generation, such as the sign of the optical orientation in

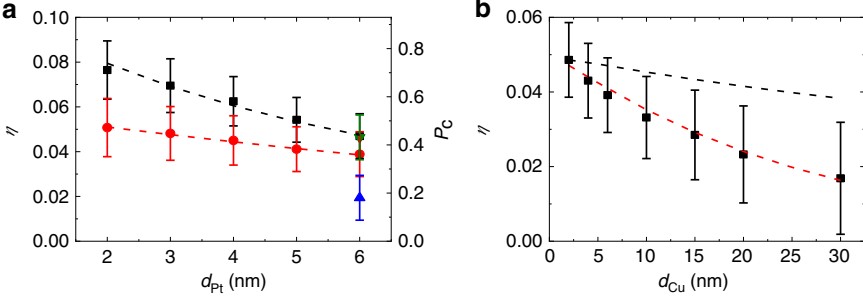

**Fig. 6 Quantum efficiency ($\eta$). a** The $\eta$ of the sap/Pt($d_{Pt}$)/Co(3)/MgO(3) (black squares) and sap/Co(3)/Pt($d_{Pt}$) (red circles) samples with a photon energy of 1.58 eV. The blue up-triangle and olive down-triangle is data with a photon energy of 1.38 and 1.73 eV, respectively, of the sap/Pt(6)/Co(3)/MgO (3) sample. The dashed lines are the spin diffusion calculation with a spin diffusion length of Pt of 6 nm (black line) and 9 nm (red line) (see Methods). The right column of **a** shows the spin-collection probability on Co ($P_c$), calculated with a spin diffusion length of Pt of 6 nm. **b** The $\eta$ of the sap/Pt(5)/Cu($d_{Cu}$)/ Co(3)/MgO(3) samples (black squares) with a photon energy of 1.58 eV. The dashed lines are the spin-diffusion calculation with a spin diffusion length of Cu of 400 nm (black line) and 25 nm (red line) (see Method). The error bar indicates the confidence interval of $\eta$ considering the uncertainty of the dynamics Kerr rotation, static Kerr rotation, and light absorption.

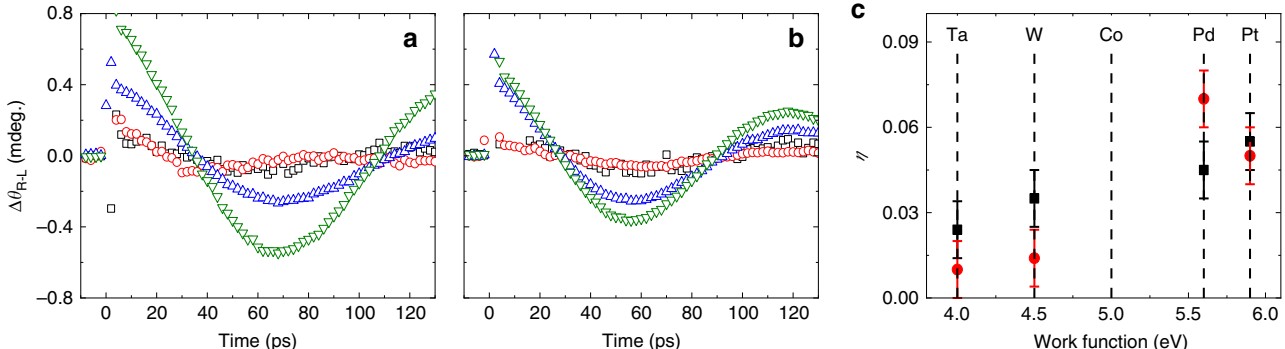

**Fig. 7 Comparison with other HMs.** The helicity-dependent Kerr rotation, difference between RCP and LCP, ($\Delta\theta_{R\text{-}L}$) of **a** the sap/HM(5)/Co(3)/MgO(3) and **b** the sap/HM(5)/Cu(2)/Co(3)/MgO(3) samples, where HM is Ta (black squares), W (red circles), Pd (blue up-triangles), and Pt (olive down-triangles) at a photon's energy of 1.58 eV. The y-axis of **b** is the same as that of **a**. **c** The quantum efficiency ($\eta$) of the sap/HM(5)/Co(3)/MgO(3) (black squares), sap/HM(5)/Cu(2)/Co(3)/MgO(3) (red circles) samples. The $\eta$ is calculated from dynamics Kerr rotation of **a** and **b**, static Kerr rotation (Supplementary Note 6), and light absorption (Supplementary Note 5). The vertical dashed lines indicate the work functions Ta, W, Co, Pd, and Pt at the (111) surface[32].

HM and the mechanism of the spin transport from HM to FM. As a final remark, we expect that the OSOT could be closely related to all-optical helicity-dependent switching observed in similar metallic heterostructures that contain HMs[33].

## Methods

**Pump-probe measurement**. For a light source, we use a Ti-sapphire oscillator (Coherent, Vision S), which produces a pulsed laser of a tunable wavelength of 690–1050 nm at a repetition rate of 80 MHz. The initial pulse width of 0.1 ps is broadened after passing through an electro-optic modulator (Conoptics, 350–160). The full-width-at-half-maximum (FWHM) at the sample is 1.1 ps and 0.2 ps for pump and probe pulses, respectively. The pump and probe are focused on the sample by a focusing lens with a diameter of the spot size of 11 μm. The pump is always incident normal to the film surface (incident angle of 0º). The incident angle of the probe is 0º for a polar MOKE and 20º for a longitudinal MOKE. The pump and probe are modulated at 10 MHz and 200 Hz, respectively, using an electro-optic modulator and a chopper. A balanced detector detects the probe reflected off the sample and converts the polarization change of the probe to the voltage signal. A lock-in amplifier detects the voltage signal that has the modulation frequency.

**Theoretical calculation of optical orientation**. With Bloch basis, the Heisenberg equation of motion for the density matrix $\rho jl$ is given by[21,34]

$$\frac{\partial \rho_{jl}(t)}{\partial t} = \frac{1}{i\hbar}\left[\epsilon_j - \epsilon_l\right]\rho_{jl} + \frac{1}{i\hbar}[\mathcal{H}^p\rho - \rho\mathcal{H}^p]_{jl} - \frac{\rho_{jl} - \rho_{jl}^0}{T_1\delta_{jl} + T_2(1-\delta_{jl})}$$

where $\epsilon_j$ is the $j$th eigenstates, $\mathcal{H}^p$ is a perturbing potential due to the light pulse, and $T_1$ and $T_2$ are phenomenological parameters for dissipative dynamics[35]. We

solve this equation within a rotating wave approximation. To be specific, we consider 6-layers-thick film of fcc-Pt piled up along the [111]-direction, and apply a circular-polarized light pulse propagating in the [111]-direction (i.e., the z-direction). We apply a continuous light with $\hbar\omega = 1.58$ eV and field strength = 0.005 V Å$^{-1}$ comparable to the experimental values).

**Spin diffusion simulation**. We calculate the diffusive transport of spins from Pt to Co in the Pt/Co and Pt/Cu/Co structure. The diffusion constants ($D$) of 220, 9500, and 250 nm$^2$ ps$^{-1}$ for Pt, Cu, and Co, respectively, are determined by $\frac{\sigma}{e^2 N_F}$, where $\sigma$ is the electrical conductivity, $e$ is the elementary charge, and $N_F$ is the electronic density of states at the Fermi level. The $\sigma$ of $6.6 \times 10^6$, $3.7 \times 10^7$, and $6.7 \times 10^6$ $\Omega^{-1}$ m$^{-1}$ for Pt, Cu, and Co, respectively, are obtained from the four-probe measurements. The $N_F$ of $1.15 \times 10^{48}$, $1.6 \times 10^{47}$, and $1.06 \times 10^{48}$ states J$^{-1}$ m$^{-3}$ for Pt, Cu, and Co, respectively, are determined by $\frac{3\gamma}{\pi^2 k_B^2}$, where $\gamma$ is the coefficient of the electronic heat capacitance[36]. The spin chemical potentials of Pt, Cu, and Co are connected at the interface with relevant spin conductances. For the Pt/Co structure, we use Re[$G_{\uparrow\downarrow}$] of $0.6 \times 10^{15}$ $\Omega^{-1}$ m$^{-1}$ at the Pt/Co interface[18]. For the Pt/Cu/Co structure, we use $\frac{G_\uparrow + G_\downarrow}{2}$ of $0.7 \times 10^{15}$ $\Omega^{-1}$ m$^{-1}$ at the Pt/Cu interface[37] and Re[$G_{\uparrow\downarrow}$] of $0.6 \times 10^{15}$ $\Omega^{-1}$ m$^{-1}$ at the Cu/Co interface[38]. With a uniform spin generation in the Pt bulk, we calculate the spin current to Co by setting the spin chemical potential of Co to be zero. Given a spin generation rate on Pt, the spin relaxation time ($\tau_s$), which is related to the spin diffusion length ($l_s$) by $l_s = \sqrt{D\tau_s}$, of Pt and Cu determines the amount of spin current to Co.

## Data availability

The data that support the findings of this study are available from the corresponding author on request.

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

## Acknowledgements

G.M.C acknowledges the National Research Foundation of Korea (NRF) grant funded by the Korea government (MSIP) (NRF-2019R1C1C1009199). K.J.L. acknowledges the National Research Foundation of Korea (NRF) grant funded by the Korea government (MSIP) (NRF-2015M3D1A1070465) and the KIST Institutional Program (project no. 2V05750). H.W.L. acknowledges the National Research Foundation of Korea (NRF) grant funded by the Korea government (MSIP) (NRF 2018R1A5A6075964). K.W.K. acknowledges the financial support from IBS (Project Code No. IBS-R024-D1). B.C.M. acknowledges the National Research Council of Science & Technology (NST) grant funded by the Korea government (MSIP) (CAP-16-01-KIST) and the KIST institutional program.

## Author contributions

G.M.C. performed the optical experiment and transport simulation. J.H.O., D.K.L., S.W. L., K.J.L., K.W.K., M.L., and H.W.L. performed the theoretical investigation. G.M.C. and B.C.M. prepared samples. All authors discussed the results and wrote the manuscript.

## Competing interests

The authors declare no competing interests.
