## [Peer Review File · Nature Communications]

Editorial Note: This manuscript has been previously reviewed at another journal that is not operating a transparent peer review scheme. This document only contains reviewer comments and rebuttal letters for versions considered at Nature Communications .

Reviewers' comments:

Reviewer #1 (Remarks to the Author):

The authors report optical spin transfer torque measurements on heavy-metal/cobalt metal layers. The experiments are on an important topic and the data is interesting. In its current form, it is not suitable for publication in Nature Communications. The manuscript requires major revisions. This manuscript does not yet meet Nature Communications requirement that the manuscript contain "all elements necessary for interpretation and replication of the results."

1) Figure 2, which reports their key experimental results, is not as clear as it should be. The axis labels on (a) (b) and (c) appear to be messed up, as well as some labels on other figures. Importantly, they don't show their longitudinal MOKE data. Instead, they just show their "conversion" of those signals into M_y in Fig. 2b. They should show the signals they are measuring, because the difference in polar and longitudinal signals is central to their claims.

2) It's not clear how the authors went from experimental signals to M_z and M_y in Fig. 2b. They include a supplemental note about this, but it describes it in very broad terms. No equations are provided that would enable someone trying to repeat the conversion on their own. For longitudinal MOKE, the polarization rotation will depend on the angle of incidence of 20 degrees. They should provide details, i.e. the relevant magneto-optical equations, that relate what is measured (voltage proportional to polarization rotation) to magneto-optical constants to M_z and M_y .

It looks more detail about the longitudinal MO effect was requested in the prior referee report, but the authors seemed to misunderstand the comment. They added a supplemental note about the polarization of the probe beam in polar MOKE. This is irrelevant because p and s polarization are equivalent in a polar configuration. They did not discuss how the angle of incidence effects signals in the longitudinal MOKE configuration, or how they account for it to get Fig. 2b.

3) Figure 2c is referred to as "amount of measured optical spin-orbit torque" but has units of Amps, not torque. They say the dashed lines are fitting with light absorption in Pt and Co. It's not clear what they mean by "with light absorption in Pt and Co."

4) On lines 145-151 they explain Fig. 2c. They say they find the "pump absorption in Pt exhibits the exactly same d_{Co} dependence as $\Delta M \cdot d_{Co}$. It would be helpful to show how the pump absorption in Pt depends on d_{Co} . The amount of light absorbed by the top Pt layer shouldn't really care how thick the Co layer once it's thicker than the optical penetration depth. In other words, since the Pt is on top, once no "reflection" occurs from interfaces below the Cobalt layer, the number of photons absorbed in Pt should stay constant. But the authors report $\Delta M \cdot d_{Co}$ steadily decreasing with Cobalt thickness. So, the conclusion that the "pump absorption in Pt exhibits exactly" the same dependence needs to be clarified.

5) The authors invoke "built-in potential" at the interface as a possible explanation for the spin-current from Pt to Co. Both prior referee reports pointed out this is a more questionable argument to make for metal-metal junctions. The authors ignored the prior request to discuss relevant literature and cite prior work on this topic. Instead, they cite a semiconductor device physics text book. Are the authors the first people to ever propose an optically induced charge current at a metal-metal interface? If not, why is there no discussion of the literature on this topic?

6) The authors describe a "charge" current that results from optical excitation. The time-scale for charge current to relax in a metal under an open circuit condition should be much shorter than their experimental time-scales, i.e. $\tau = \epsilon/\sigma \sim 1e-18$ seconds [PRB 95 014402 (2017)]. Here ϵ is the permittivity and σ is the conductivity. So their conclusion that the pump pulse generates a 1.1 ps charge current that can traverse 25 nm is hard to understand.

They don't provide enough details to understand what exactly their "charge transport simulation" is. They just say they used a "SPICE simulation. This is another example where the manuscript doesn't contain all elements necessary to replicate the results.

Reviewer #2 (Remarks to the Author):

The submitted paper reports on the helicity-dependent optical spin-injection experiments in structures containing heavy metal (HM) on cobalt. I have reviewed the closely related manuscript in 2017, when the authors submitted it to Nature Nanotechnology. At this time, I had several - rather serious - doubts/questions about the physical mechanisms that were proposed as explanations of the performed experiments. In the current version of the manuscript, the authors have addressed all my previous experimental and theoretical concerns. The resulting paper is, at least in my eyes, very interesting and I have no doubts that it will attract a considerable attention of scientific community after its publication. At present, I have only one request, which is detailed below, that I would like the authors to address in the paper (or in the modified Supplementary information). After this, I recommend the publication of the paper in Nature Communication.

It is a very interesting and non-trivial result that the optical spin-injection efficiency has the same sign for different HMs, which have different work functions - see Fig. 5(c). However, as there are several experimental inputs that could affect the determined sign, I would find it very useful if the authors could provide also the "raw" experimental data from which this figure was derived. In particular, it would be nice to see the time-resolved magneto-optical data [analogous to Fig. 2(a)] and static Kerr rotations [analogous to Fig. S4] measured in samples with various HMs.

Response letter to Referee's comments

“Optical spin-orbit torque in heavy metal/ferromagnet heterostructures” by G. M. Choi *et al.*

We thank both referees for their constructive comments. Below, we summarize point-by-point responses to the referee's comments and suggestions. The responses are marked in blue. The corresponding corrections are incorporated in the revised manuscript.

Reviewers' comments:

Reviewer #1 (Remarks to the Author):

The authors report optical spin transfer torque measurements on heavy-metal/cobalt metal layers. The experiments are on an important topic and the data is interesting. In its current form, it is not suitable for publication in Nature Communications. The manuscript requires major revisions. This manuscript does not yet meet Nature Communications requirement that the manuscript contain “all elements necessary for interpretation and replication of the results.”

1) Figure 2, which reports their key experimental results, is not as clear as it should be. The axis labels on (a) (b) and (c) appear to be messed up, as well as some labels on other figures. Importantly, they don't show their longitudinal MOKE data. Instead, they just show their “conversion” of those signals into M_y in Fig. 2b. They should show the signals they are measuring, because the difference in polar and longitudinal signals is central to their claims.

→ In Fig. 2 (b), we show the normalized Kerr rotation to emphasize the difference of dynamics between M_z and M_y components. The process to measure M_y dynamics will be discussed in our response to the reviewer's next comment. To avoid confusion with axis in Fig 2, we move the plot of $\Delta M \cdot d_{Co}$, which is a measure of $\int J_s dt$, where J_s is the spin (magnetic moment) current per area to Co, as a separate figure (Fig. 3) in the revised manuscript.

2) It's not clear how the authors went from experimental signals to M_z and M_y in Fig. 2b. They include a supplemental note about this, but it describes it in very broad terms. No equations are provided that would enable someone trying to repeat the conversion on their own. For longitudinal MOKE, the polarization rotation will depend on the angle of incidence of 20 degrees. They should provide details, i.e. the relevant magneto-optical equations, that relate what is measured (voltage proportional to polarization rotation) to magneto-optical constants to M_z and M_y .

It looks more detail about the longitudinal MO effect was requested in the prior referee report, but the authors seemed to misunderstand the comment. They added a supplemental note about the polarization of the probe beam in polar MOKE. This is irrelevant because p and s polarization are equivalent in a polar configuration. They did not discuss how the angle of incidence effects signals in the longitudinal MOKE configuration, or how they account for it to get Fig. 2b.

→ We check the probe polarization dependence in polar MOKE geometry to confirm that a quadratic Kerr rotation, which can have the M_y information even at normal incidence angle, is not significant in our samples. To measure the M_y dynamics, we inject the probe beam to the sample with an oblique angle, θ_0 , in the y - z plane (Fig. S3 of Supplementary Note). In this

geometry, the measured Kerr rotation is a mixture of the polar Kerr rotation (θ_K^p), which is due to M_z dynamics, and the longitudinal Kerr rotation (θ_K^l), which is due to M_y dynamics. When the sample consists of a single magnetic layer whose thickness is much thicker than the optical penetration depth, the ratio between the θ_K^l and θ_K^p has a simple relationship of $\frac{\theta_K^l}{\theta_K^p} = \tan \theta_1$, where θ_1 is the angle of refraction determined by θ_0 and refractive index of the magnetic layer [*J. Appl. Phys.* **84**, 541 (1998)]. When the sample consists of multilayers with a thickness thinner than the optical penetration depth, there is no simple relation between θ_K^l and θ_K^p . To distinguish θ_K^l and θ_K^p , we measure the Kerr rotation with an applied field of $+x$ and $-x$ directions. The sign of the M_z dynamics does not change with $+x$ and $-x$ directions of an applied field, but the sign of the M_y dynamics does. Therefore, the M_z and M_y dynamics can be obtained from the even and odd parts of the raw data (Fig. S4 of Supplementary Note). Figure 2 (b) of the main text is obtained by normalizing the plot of Fig. S4 (c). We include this discussion in the Supplementary Note 3.

3) Figure 2c is referred to as “amount of measured optical spin-orbit torque” but has units of Amps, not torque. They say the dashed lines are fitting with light absorption in Pt and Co. It’s not clear what they mean by “with light absorption in Pt and Co.”

→ In Fig. 2 (c) of the previous submission, we show the magnitude of optical spin-orbit torque in the form of $\Delta M \cdot d_{Co}$, which has a unit of “A”. The $\Delta M \cdot d_{Co}$ is a measure of $\int J_s dt$, where J_s is the spin (magnetic moment) current per area to Co. J_s produces a torque on magnetization of Co. In our experiment, J_s appear as a short pulse, and the magnitude of magnetization dynamics is determined by $\int J_s dt$. We include the explanation of $\Delta M \cdot d_{Co}$ in line 151~154 and redraw the result of $\Delta M \cdot d_{Co}$ and light absorption calculation as Fig. 3 of

the revised manuscript.

For the light absorption, we calculate the Poynting vector of pump light as it passes through the sample using a transfer matrix method. The decrease of the Poynting vector by each layer represents the amount of light absorption. Given the incident pump fluence, the amount of light absorption in Pt and Co of the sap/Co(x)/Pt(2) structure, where the pump is incident on the sapphire substrate side, depends on the thickness of Co. As the Co thickness increases, the light absorption of Co increases and then saturates, whereas that of Pt decreases continually. We explain how to calculate the light absorption in Supplementary Note 5 and include the calculation of light absorption as an inset of Fig. 3 of the revised manuscript.

4) On lines 145-151 they explain Fig. 2c. They say they find the “pump absorption in Pt exhibits the exactly same d_{Co} dependence as $\Delta M \cdot d_{\text{Co}}$. It would be helpful to show how the pump absorption in Pt depends on d_{Co} . The amount of light absorbed by the top Pt layer shouldn't really care how thick the Co layer once it's thicker than the optical penetration depth. In other words, since the Pt is on top, once no “reflection” occurs from interfaces below the Cobalt layer, the number of photons absorbed in Pt should stay constant. But the authors report $\Delta M \cdot d_{\text{Co}}$ steadily decreasing with Cobalt thickness. So, the conclusion that the “pump absorption in Pt exhibits exactly” the same dependence needs to be clarified.

→ The amount of light absorption in Pt of the sap/Co(x)/Pt(2) structure, where the pump is incident on the sapphire substrate side, depends on the thickness of Co since the pump should pass the Co layer before it reaches the Pt layer. As the Co thickness increases, the light absorption of Co increases then saturates, and that of Pt decreases continuously. When the Co thickness becomes much thicker than the optical penetration depth, there will be no light absorption of Pt as light cannot reach Pt. We include the calculation of light absorption as an inset of Fig. 3 of the revised manuscript.

We expect the reviewer misunderstood about the location of the pump pulse. The pump pulse is incident on the sapphire substrate side of the samples: for sap/Pt/Co or sap/Pt/Cu/Co samples, the pump passes the substrate layer and then the Pt layer; for sap/Co/Pt samples, the pump passes the Co layer and then the Pt layer. We included this explanation in line 103~106 of the revised manuscript.

We also note that the quantum efficiency of the sap/Co(3)/Pt(x) samples is somewhat smaller than that of the sap/Pt(x)/Co(3)/MgO(3) samples probably because different morphology at the Pt/Co interface affects the spin transport efficiency from Pt to Co. We include this explanation in line 241~243 and show the quantum efficiency in Fig. 6 (a) of the revised manuscript.

5) The authors invoke “built-in potential” at the interface as a possible explanation for the spin-current from Pt to Co. Both prior referee reports pointed out this is a more questionable argument to make for metal-metal junctions. The authors ignored the prior request to discuss relevant literature and cite prior work on this topic. Instead, they cite a semiconductor device physics text book. Are the authors the first people to ever propose an optically induced charge current at a metal-metal interface? If not, why is there no discussion of the literature on this topic?

→ We argue that the existence of built-in potential (or E -field) at interfaces is general regardless of the band structure of materials as charge transfer occurs at the interface between materials with different work functions. We quote a theoretical paper that shows the built-in potential of composite metallic interfaces of a few tens of meV [*J. Phys. Chem. Lett.* **3**, 818 (2012)]. We also quote three papers about Rashba spin splitting at metallic interfaces, for which an E -field at the interfaces is a key requirement [*Phys. Rev. Lett.* **77**, 3419-3422 (1996), *Phys. Rev. B* **90**, 235422 (2014), and *Phys. Rev. B* **93**, 174421 (2016)]. However, we admit

that the magnitude of the built-in potential in a metallic junction would be much smaller than that of a semiconductor junction. Especially, the E -field should be strong enough to induce photocurrent. We explicitly stated this assumption for the photocurrent in line 267~272 and moved the detailed analysis for the charge transport to Supplementary Note 7. We remove the word “built-in potential” as it is rarely used for metallic junctions. Instead we use the word “ E -field” as it is often used to describe the Rashba effect.

6) The authors describe a “charge” current that results from optical excitation. The time-scale for charge current to relax in a metal under an open circuit condition should be much shorter than their experimental time-scales, i.e. $\tau = \epsilon/\sigma \sim 1\text{e-}18$ seconds [PRB 95 014402 (2017)]. Here ϵ is the permittivity and σ is the conductivity. So their conclusion that the pump pulse generates a 1.1 ps charge current that can traverse 25 nm is hard to understand. They don’t provide enough details to understand what exactly their “charge transport simulation” is. They just say they used a “SPICE simulation. This is another example where the manuscript doesn’t contain all elements necessary to replicate the results.

→ Assuming that the E -field at the Pt/Cu interface is strong enough, we calculate the photo-induced charge transport in the Pt(5)/Cu(x)/Co(3) structure using SPICE simulation. Since the charge current flows along the out-of-plane direction in metallic multilayers on top of the insulating substrate, it is subject to the open circuit situation. With a homogeneous metal at the open circuit, any forward charge current (J_c) will be quickly canceled out by the backward current. However, when an electric field at the interface is strong enough, the backward current should be blocked. The circuit diagram for the simulation is shown in Fig. S7 (a) of Supplementary Note. The photocurrent is modeled as a current source that generates a forward current (J_c). J_c at the Pt/Cu interface is a Gaussian pulse with FWHM of 1.1 ps, and its magnitude is set by $\int J_c dt = \frac{q}{\hbar\omega} F_{\text{in}} A_{\text{Pt}}$. The effect of the electric field at the

Pt/Cu interface is modeled as a diode, which blocks a backward current. The threshold voltage of the diode is set to 0.7 V. The capacitances of Cu and Co are determined from the electronic density of states at Fermi level (N_F) and thicknesses (d) as $C = N_F \cdot d$. The resistances of Cu and Co are determined from bulk electric resistivity and interfacial resistance, but as long as small enough they do not affect the simulation result. The charge accumulation ($Q_c = \int J_c dt$) at the Co capacitance decreases with increasing Cu thickness with an exponential length scale of 20~30 nm (Fig. S7 (b)). We also calculate the charge accumulation on Co with a relative ratio of the electronic capacitance of Co and Cu. When we ignore the resistance effect and assume no backward current, charge accumulation Co (Q_{Co}) will be determined by

$$Q_{Co} = Q_0 \frac{N_{F,Co} d_{Co}}{N_{F,Cu} d_{Cu} + N_{F,Co} d_{Co}}, \quad (1)$$

where Q_0 is the time integration of J_c . When d_{Co} is fixed, Q_{Co} decrease with increasing d_{Cu} , and the calculated length scale of the decaying is 20~30 nm, which is the same as the SPICE simulation (Fig. S7 (b)).

The length scale of 20~30 nm of the charge current at open circuit might look too long considering the short timescale for the charge relaxation $\tau_{dr} = \epsilon/\sigma$, where ϵ is permittivity and σ is electrical conductivity [*Phys. Rev. B* 95, 014402 (2017)]. However, we would like to point out that having short τ_{dr} does not imply a short characteristic length scale of charge current transport. To demonstrate this point, one begins with the local charge conservation equation,

$$\frac{\partial \rho}{\partial t} + \nabla \cdot \vec{j} = 0, \quad (2)$$

where ρ is the charge density and \vec{j} is the charge current density. One then combines this equation with the Ohm's law $\vec{j} = \sigma \vec{E}$ and the Gauss law $\nabla \cdot \vec{E} = \rho/\epsilon$ to obtain the

approximate equation,

$$\frac{\partial \rho}{\partial t} + \frac{\rho}{\tau_{dr}} = 0. \quad (3)$$

The solution of this approximate equation is given by $\rho(\vec{r}, t) = \rho(\vec{r}, t = 0)\exp(-t/\tau_{dr})$, which implies that the initial charge density at $t = 0$ decays exponentially fast in time with the characteristic time scale τ_{dr} . We are now ready to discuss the length scale. For concreteness, one considers a metallic thin film (shown below) and assumes $\rho(\vec{r}, t = 0)$ to be spatially homogeneous within the film, $\rho(\vec{r}, t = 0) = \rho_0$ (this assumption is not crucial, however).

For this film geometry, one then uses the Gauss law to calculate the electric field \vec{E} from the above solution $\rho(\vec{r}, t)$ to obtain $\vec{E}(\vec{r}, t) = \hat{z} \left(\frac{z\rho_0}{\epsilon} \right) \exp\left(-\frac{t}{\tau_{dr}}\right) + \text{constant}$. Together with the Ohm's law, one obtains

$$\vec{j}(\vec{r}, t) = \hat{z} z \frac{\rho_0}{\tau_{dr}} \exp\left(-\frac{t}{\tau_{dr}}\right) + \text{constant}, \quad (4)$$

where z is the coordinate along the thickness direction and \hat{z} is the unit vector along the z direction. Note that $\vec{j}(\vec{r}, t)$ does grow in space (proportional to z), implying that the characteristic length scale is *long-ranged*. This long-rangedness persists even for spatially localized $\rho(\vec{r}, t = 0)$, say, $\rho(\vec{r}, t = 0) = A\delta(z)$. For this initial condition, one obtains

$$\vec{j}(\vec{r}, t) = \hat{z} \text{sgn}(z) \frac{A}{2\tau_{dr}} \exp\left(-\frac{t}{\tau_{dr}}\right), \quad (5)$$

where $\text{sgn}(z)$ is the Heaviside step function. Note that $\vec{j}(\vec{r}, t)$ for this choice of

$\rho(\vec{r}, t = 0)$ is also spatially long-ranged. This result for $\vec{J}(\vec{r}, t)$ actually implies that when a local charge density $\rho(\vec{r}, t = 0) = A\delta(z)$ (assume $A > 0$) is introduced by an external perturbation at time $t = 0$ at the mid-plane ($z = 0$) of the film, the initial charge relaxes by generating $\vec{J}(\vec{r}, t)$ [toward $+\hat{z}$ direction for $z > 0$ and toward $-\hat{z}$ for $z < 0$] over the *entire* thickness of the film. This charge current density is maintained for the time of the order of τ_{dr} and after this time, the initial charge is entirely transported to the upper and lower surfaces of the film and piled up there due to the open boundary condition. This verifies that even when the characteristic time scale τ_{dr} is very short, $\vec{J}(\vec{r}, t)$ within this time interval can be extended over long distances. In the approximation presented above, the characteristic length scale of $\vec{J}(\vec{r}, t)$ is bounded only by the film thickness. We included this discussion in Supplementary Note 7.

Reviewer #2 (Remarks to the Author):

The submitted paper reports on the helicity-dependent optical spin-injection experiments in structures containing heavy metal (HM) on cobalt. I have reviewed the closely related manuscript in 2017, when the authors submitted it to Nature Nanotechnology. At this time, I had several - rather serious - doubts/questions about the physical mechanisms that were proposed as explanations of the performed experiments. In the current version of the manuscript, the authors have addressed all my previous experimental and theoretical concerns. The resulting paper is, at least in my eyes, very interesting and I have no doubts that it will attract a considerable attention of scientific community after its publication. At present, I have only one request, which is detailed below, that I would like the authors to address in the paper (or in the modified Supplementary information). After this, I recommend the

publication of the paper in Nature Communication.

It is a very interesting and non-trivial result that the optical spin-injection efficiency has the same sign for different HMs, which have different work functions – see Fig. 5(c). However, as there are several experimental inputs that could affect the determined sign, I would find it very useful if the authors could provide also the "raw" experimental data from which this figure was derived. In particular, it would be nice to see the time-resolved magneto-optical data [analogous to Fig. 2(a)] and static Kerr rotations [analogous to Fig. S4] measured in samples with various HMs.

→ We include the raw data of the time-resolved Kerr rotation with HM of Ta, W, Pd, and Pt in Fig. 7 of the revised manuscript and those of the static Kerr rotation in Fig. S6 of the Supplementary Note. We also include the calculation result of the light absorption of HM in Supplementary Table S1.

REVIEWERS' COMMENTS:

Reviewer #1 (Remarks to the Author):

The manuscript contains interesting and novel experimental results regarding the response of magnetic order in ferromagnetic orders to circularly polarized light. The results will be of interest to a broad audience. The authors have done a thorough job responding to my comments by adding significant detail describing how the experiments and analysis was conducted. I recommend for publication in Nature Communications.

Reviewer #2 (Remarks to the Author):

In my eyes, the authors addressed reasonably well all the questions/comments of the referees. Therefore, I recommend the publication of the paper in Nature Communication.

Response letter to Referee's comments

“Optical spin-orbit torque in heavy metal-ferromagnet heterostructures” by G. M. Choi *et al.*

We thank both referees for their constructive comments. Since both reviewers do not ask any further work, we do not change the main contents of the manuscript.

Reviewers' comments:

Reviewer #1 (Remarks to the Author):

The manuscript contains interesting and novel experimental results regarding the response of magnetic order in ferromagnetic orders to circularly polarized light. The results will be of interest to a broad audience. The authors have done a thorough job responding to my comments by adding significant detail describing how the experiments and analysis was conducted. I recommend for publication in Nature Communications.

Reviewer #2 (Remarks to the Author):

In my eyes, the authors addressed reasonably well all the questions/comments of the referees. Therefore, I recommend the publication of the paper in Nature Communication.